# Effects of Heat Input on Weld Microstructure and Properties in Keyhole TIG Welding of Invar 36 Alloy

**DOI:** 10.3390/ma16103692

**Published:** 2023-05-12

**Authors:** Hongbing Liu, Shanhui Lv, Yang Xuan, João Pedro Oliveira, Norbert Schell, Jiajia Shen, Jingyu Deng, Yuhua Wang, Jin Yang

**Affiliations:** 1School of Materials Engineering, Shanghai University of Engineering Science, Shanghai 201620, China; lhongbing@163.com (H.L.);; 2Shanghai Collaborative Innovation Center of Intelligent Manufacturing Robot Technology for Large Components, Shanghai 201620, China; 3Shanghai No. 1 Machine Tool Works Co., Ltd., Shanghai 201620, China; 4UNIDEMI, Departamento de Engenharia Mecânica e Industrial, Faculdade de Ciências e Tecnologia, Universidade NOVA de Lisboa, 2829-516 Caparica, Portugal; 5CENIMAT/I3N, Department of Materials Science, NOVA School of Science and Technology, Universidade NOVA de Lisboa, 2829-516 Caparica, Portugal; 6Institute of Materials Physics, Helmholtz-Zentrum Hereon, Max-Planck-Str. 1, 21502 Geesthacht, Germany; 7Shanghai Aircraft Manufacturing Co., Ltd., Commercial Aircraft Corporation of China, Shanghai 200120, China

**Keywords:** K-TIG welding, Invar 36 alloy, heat input, mechanical properties, synchrotron X-ray diffraction

## Abstract

The Invar alloy is widely used for aircraft wing mould manufacturing. In this work, keyhole-tungsten inert gas (K-TIG) butt welding was used to join 10 mm thick Invar 36 alloy plates. The effect of heat input on the microstructure, morphology and mechanical properties was studied by using scanning electron microscopy, high energy synchrotron X-ray diffraction, microhardness mapping, tensile and impact testing. It was shown that regardless of the selected heat input, the material was solely composed of austenite, although the grain size changed significantly. The change in heat input also led to texture changes in the fusion zone, as qualitatively determined with synchrotron radiation. With increases in heat input, the impact properties of the welded joints decreased. The coefficient of thermal expansion of the joints was measured, which demonstrated that the current process is suitable for aerospace applications.

## 1. Introduction

Invar alloys have been widely used in large moulds in the aerospace industry [1] and liquefied natural gas tankers due to their extremely low coefficient of thermal expansion (CTE). The low CTE of these alloys occurs due to the Invar effect, i.e., when the atomic concentration of Ni in Fe–Ni alloy is about 36%, the alloy shows the lowest CTE [2]. Invar alloys are austenitic alloys, and their low coefficient of thermal expansion is easily affected by their composition. The austenite structure with a face-centered lattice structure is also a solid solution formed by nickel dissolved in γ-Fe. Therefore, when welding is used, cracking, porosity and CTE mismatch with the base metal (BM) are prone to occur.

Metal inert gas (MIG) and tungsten inert gas (TIG) are traditional welding methods for joining of Invar 36 alloys [3,4,5]. More recently, some advanced welding techniques have also been widely used for joining of Invar 36 alloys [6], and these include laser welding, laser–MIG hybrid welding and friction stir welding (FSW). However, each process has its own advantages and disadvantages. For example, Zhao et al. [7] obtained Invar 36 alloy welds with excellent properties and CTE using laser welding, but the method was not suitable for middle-thickness Invar 36 alloy plates. Li et al. [8] used laser–MIG hybrid welding to join Invar 36 alloy plates, and the joints possessed excellent mechanical performance, but the CTE was significantly higher than that of the BM. Jasthi et al. [9] performed FSW with an Invar 36 alloy, and this process eliminated porosity, cracking, and other severe problems which often occur during fusion welding. However, one the major drawbacks of FSW is its low process flexibility. Compared with TIG welding and MIG welding, K-TIG welding has higher penetration, good weld forming quality and high efficiency.

Keyhole TIG (K-TIG) welding, an upgraded version of TIG, is a new deep penetration welding technology that encompasses a significant arc force. Furthermore, it is capable of realizing single-pass welding of middle-thick plates without the need for filler metal and grooves [10]. In addition, K-TIG allows a higher welding speed and has wider process parameters than plasma arc welding, which is of extreme importance for industrial applications [11]. Due to the aforementioned benefits, this welding technology has been used in the welding of middle thick plates, such as zirconium [12], low carbon steel [13], titanium [14] and stainless steel [15], as well as for dissimilar joints [16]. In combination with auxiliary methods, K-TIG can also be used for high thermal conductivity materials [17], armour steel [18] and underwater welding of duplex stainless steel [19]. Furthermore, Liu et al. [20,21] found that the size, shape, and position of the keyhole outlet is highly dynamic. Because of the large penetration of K-TIG welding, the grain size of the weld is coarser. Large penetration is both an advantage and a disadvantage for K-TIG welding. Due to the large heat input in K-TIG welding, the performance of the weld is negatively affected to some extent.

In our previous study, K-TIG welding was used for welding of Invar 36 alloy [22]. However, the change in morphology, microstructure and mechanical properties of the weld joints obtained with different welding parameters was not comprehensively discussed. The relations among heat input and microstructure evolution are far from being completely established. Thus, in the present study, K-TIG welding with different heat inputs was applied to butt joining of 10 mm thick Invar 36 alloy plates. The weld appearance and microstructure characteristics were analysed. The weld properties, including microhardness, tensile strength, and impact properties with different heat inputs, were systematically investigated.

## 2. Materials and Experiment Methods

### 2.1. Materials

Invar 36 alloy with a thickness of 10 mm was used. The composition and properties of the Invar 36 alloy are presented in Table 1. The Invar 36 alloy plates were cut to 100 mm × 150 mm × 10 mm specimens before welding.

### 2.2. Welding Process

Prior to welding, the plates were cleaned by mechanical grinding and acetone. Autogenous single-pass welding was performed. The equipment (shown in Figure 1) for the K-TIG welding was provided by Shanghai Duomu Industrial Co., Ltd. The welding parameters are detailed in Table 2. The heat input was calculated using the following equation:(1)q=ƞUI/v
where *q* is the welding heat input (J/mm), *U* is the welding voltage (V), *I* is the welding current (A), *v* is the welding speed (mm/s), and *ƞ* is the process efficiency, which was set to 0.9.

### 2.3. Experiment Methods

Grinding of the welded samples for metallography was performed with SiC papers with grades from 180# to 1200#. Then, the samples were polished with 0.5 μm diamond suspension. Finally, the specimens were chemically etched in 40 mL of aqua regia for 20 s. The microstructure was characterized using an 4XCJZ optical microscope and a S-3400N scanning electron microscope (SEM). The grain size mentioned in this paper was measured by imageJ software. The average grain size was measured multiple times within the same region of the weld to ensure an adequate statistical sample. High-energy synchrotron X-ray radiation was used to scan the sample from one side of the base material to the other side, passing by both the heat affected zones (HAZ) and fusion zones (FZ). Experiments were performed at the P07 HEMS beamline of DESY, using a beam energy of 87 keV and a 2D Perkin Elmer fast detector. Data processing was performed using an in-house developed python routine available at CENIMAT/I3N. Tensile testing was performed on an automatic drawing machine (WE-60) using a displacement speed of 1 mm/min at room temperature. The specimen’s dimensions were defined according to the ASTM E8-13a standard and are detailed in Figure 2a [23]. According to ASTM E23-18 [24], the Charpy impact test was performed with a pre-added V-notch placed in the HAZ and the FZ as shown in Figure 2b,c. X-ray radiographs were used to detect defects such as porosity and cracks in the joints.

## 3. Results and Discussion

### 3.1. Macrostructure

Figure 3a–c present the cross-sections of the welds obtained with different heat inputs (2895.3, 3102.1 and 3308.9 J/mm). All the welds had full penetration, but the weld appearance changed significantly. The width of weld at the half-thickness of the joint and the width of the weld root were used to represent the joint size, where both increased with the increase in the heat input. The large heat input increased the weld pool size, which increased both the root and weld widths.

With the increase in heat input, the reinforcement of the weld decreased gradually, while the depth of the depression (undercut) observed at the face of the welded joint increased gradually, as shown in Figure 3a–c. Figure 4 shows a schematic diagram of the weld pool under different heat inputs. When the heat input was low, melting of the material at the bottom of the melt pool was not complete, which led to a large bending of the front wall of the pool. A large amount of plasma was discharged upward to squeeze the molten pool, which contributed to the formation of the observed reinforcement. The increase in this reinforcement may affect the tensile properties and fracture of the joint. The heat input changes the trajectory of the plasma jet in the orifice channel, resulting in the formation of a hump in the center of the weld. This intensifies the formation of edge biting, which also plays an important role in the formation of edge biting defects. Both bite defects and the root fusion boundary can be used as stress concentration points, which have a significant influence on the joint’s fracture mode and tensile properties [25].

### 3.2. Microstructure

Figure 3d–f detail the microstructure of the FZ near the HAZ obtained from magnifications of Figure 3a–c. The BM, HAZ and FZ contained fine equiaxed austenite grains, coarse equiaxed austenite grains and columnar austenite grains, respectively.

With the increase in heat input, the grain size and area of the HAZ increased significantly, as shown in Figure 5a. The large heat input led to overheating, explaining the increase in the HAZ area. The higher the temperature and residence time, the easier it is for atomic diffusion to occur, which facilitates grain growth. Typically, the HAZ is considered as a weak zone of the welded joint and may have significant influence on the resulting mechanical and functional properties.

The grain structure in the FZ was divided into upper and lower parts according to different growth directions, as detailed in Figure 3. With the increase in heat input, the grain size widths of the upper and lower parts increased, as shown in Figure 5b. Large heat input reduced the non-uniform nucleation rate and prolonged the residence time of the weld pool at high temperatures, which provided sufficient time for the growth of grains in the FZ. As is shown later, the grain size is related to the impact properties, tensile properties and hardness of the joint.

The composition results obtained in the FZ (see Spectrum 1–3 in Figure 3d–f) are detailed in Figure 6. The contents of Fe and Ni changed only slightly, indicating that the heat input had little influence on material composition.

### 3.3. Spatially Resolved Synchrotron X-ray Diffraction

Figure 7 details the spatially resolved synchrotron X-ray diffraction analysis scanning the joint from the BM to the FZ. A single-phase FCC crystal structure is preserved all across the welded joint, indicating that the selected weld heat inputs were not conducive to changes in the existing phase structure in both the HAZ and FZ [26]. However, the diffracted intensity varies between the BM, HAZ and FZ, and this is related to changes in both the grain size and texture of the welded joints.

### 3.4. Microhardness

Microhardness curves across the welds with different heat inputs were investigated as shown in Figure 8. The position of the hardness test point is marked by the weld in the upper left corner of Figure 8. The hardness of the welded joints with heat inputs of 2895.3 and 3102.1 J/mm were similar, while the hardness of the welded joint with largest heat input of 3308.9 J/mm decreased slightly. This was mainly correlated with the change in grain size in both the HAZ and FZ. The welded joint with a heat input of 3308.9 J/mm had the largest grain sizes in these regions (refer to Figure 5), leading to a reduction in microhardness. In addition, the dimensions of the HAZ and FZ were the largest, as also evidenced by these microhardness measurements.

### 3.5. Tensile Testing

Figure 9 depicts the tensile properties of the welded joints under different heat inputs. The tensile strength and joint elongation of the welded joint with different heat inputs showed only minor changes. The tensile strength with different heat inputs was 438, 427 and 431 MPa, respectively. The joint elongation with different heat inputs was 29.2, 31 and 30.8%, respectively. All the welded joints maintained excellent mechanical properties with only a slight reduction compared with the BM (441 MPa) due to changes in grain size and joint shape. Although there were changes in the microstructure and joint shape of the welded joints, as detailed in Figure 3, these were not sufficient to significantly modify the macroscopic tensile response of the welded joints obtained under distinct heat inputs.

Figure 10 shows the fracture locations of the K-TIG welds with different heat inputs. Similar fracture paths were observed in all welded joints. The fracture of the welds first initiated at the root of the weld due to stress concentration, and then the crack propagated stably towards the upper part of the FZ. The elongation, Luders bands and neck contraction of the FZ were clearly observed, as shown in Figure 10d. Since the material has an FCC structure, the deformation of each grain could not be coordinated at grain boundaries because of the hindrance of grain boundaries, resulting in uneven deformation of each grain. The grain coarsening of Invar 36 welds was significant, so a macroscopic Luders zone appeared [27]. A cross-sectional view of the fractured specimen is shown in Figure 10e. The lower part of the fracture first cracked under tensile stress, forming a fibrous zone. Then the upper part of the fracture formed under the action of a shear stress, forming a shear lip zone. The fiber region is generally the fracture source region. The shear lip is always found at the edge of the fracture and has an angle of about 45° with the surface of the component. It is formed by shear tearing under the condition of plane stress, and the surface of the shear lip is relatively smooth.

### 3.6. Charpy Impact Testing

The impact properties of K-TIG welds with different heat inputs are detailed in Table 3. The energy absorbed was 179.7, 161.8 and 151.8 J for the 2895.3, 3102.1 and 3308.9 J/mm joints, respectively. As for the HAZ, the energy absorbed until fracture was 172.8, 157.2 and 150.5 J, for the 2895.3, 3102.1 and 3308.9 J/mm joints, respectively. Compared with the BM (202.7 J), the impact properties of welded joints were slightly reduced. Moreover, with increased heat input, the impact properties of both the HAZ and FZ decreased. The grain coarsening of both the HAZ and FZ is the primary reason for the decrease in the impact properties following the Hall–Petch effect. As the heat input increased, the grain size of the HAZ and FZ increased (refer to Figure 5); thus, the impact properties of the joint decreased accordingly. The impact properties of the FZ were generally higher than those of the HAZ. The HAZ was the connection between the liquid weld pool and the solid BM, which led uneven distribution of microstructure and stress on both sides. Under the impact load, the uneven stress distribution of the notch significantly reduced the impact energy of the HAZ.

Table 3 shows the macroscopic morphology of the impact fracture. A significant amount of deformation occurred in all samples, indicating that the welded joints had good plasticity. Figure 11 and Figure 12 detail the SEM images of the impact fracture surfaces. A large amount of plastic deformation, dimple areas and shear lip areas appeared in the fracture surface, which was characteristic of ductile fracture. The fracture surfaces for the different heat inputs were similar and were divided into shear lip zone, initial fracture zone, dimple fracture zone and unfractured zone. The dimple fracture zone was mainly composed of secondary fiber zone, which was formed by crack propagation into the compressive stress zone [27]. 

Prominent lacerated dimples on the fracture surface were evident, as shown in the enlarged view of Figure 11 and Figure 12. The SEM morphology with different heat inputs was similar but the size of the dimples in the FZ was smaller than that of the HAZ. The size of dimples was not only related to the impact toughness of the material, but also to its strain hardening index [28]. The coarse austenite grains had a higher deformation hardening index, so it is difficult to shrink the neck, resulting in smaller and shallower fracture dimples [28,29]. The grain size of the FZ was significantly larger than that of the HAZ, indicating a smaller dimple size in the FZ.

### 3.7. Comparison of Mechanical Properties

Table 4 describes the Invar 36 alloy K-TIG welding properties compared with other welding processes. As can be seen from the table, the material elongation after K-TIG welding is higher, the hardness is lower, and the ultimate tensile strength compared with the base material is also higher than the other welding techniques listed. This shows that K-TIG is suitable for successful welding of Invar 36 alloys.

### 3.8. Coefficient of Thermal Expansion Test

The CTE mismatch is one of the main problems associated with welding of Invar 36 alloy. Therefore, it was necessary to determine the CTE of both the BM and the weld. The weld joints obtained with a heat input of 3102.1 J/mm were used for the CTE testing. The measured expansion values of both the BM and weld are shown in Figure 13a, and the CTE values are shown in Figure 13b. These were calculated by the following formula:(2)$$[CTE{]}_{Ti}=[\frac{1}{∆T}\frac{∆L}{L_{0}}{]}_{Ti}=\frac{1}{∆T}[(\frac{∆L}{L_{0}}{{)}_{m}+A]}_{Ti}$$
where the unit of CTE is (μm/m).

*L*_0_ is the initial length of the sample at room temperature,∆*L* is the difference of the expansion value of the sample at a certain temperature, ∆*T* is the temperature difference,d*L*/d*T* is the instantaneous expansion value at a certain temperature point,*A_Ti_* is the calibration constant used to eliminate the expansion deviation between the sample and the test fixture used in the measurement process.

Both the dimension change and CTE curves of the BM and weld were almost superimposed on each other. In general, the expansion value increased with the increase in temperature, as shown in Figure 13a. The CTE gradually increased with the increase in temperature and then decreased when the temperature was above 456 °C (Figure 13b). The curve associated with the dimension change was divided into three parts: low thermal expansion zone, transition zone, and high thermal expansion zone [29]. In the low expansion zone, the expansion value of the BM and weld increased slowly, and the difference in CTE was small. In the transition zone, the expansion value and CTE of the weld and BM increased rapidly. In the high thermal expansion zone, the expansion value maintained a high growth rate and the CTE remained at a high value. Therefore, it is inferred that the Curie temperature points of the weld and BM were basically unchanged, which further confirms that the K-TIG welding process has a very limited impact on the low expansion characteristic of Invar alloys [8]. The nickel content is one of the main factors affecting the CTE of Invar alloys [2]. The main reason that the CTE of the weld did not mismatch that of the BM, was related to the preservation of the Ni content in the FZ [22].

### 3.9. X-ray Non-Destructive Test

The quality of the K-TIG welded specimens was further assessed using non-destructive testing. The weld joints obtained with a heat input of 3102.1 J/mm were used for testing, as shown in Figure 14. Radiography was used to identify and analyse potential defects within the welded joint. It was observed that no defects such as pores and cracks existed. This suggests that K-TIG welding of Invar 36 alloy effectively solved the problems of porosity and cracking typically found in laser–MIG hybrid welding and multi-layer multi-pass welding of Invar 36 alloy [33,34].

The large heat input of K-TIG welding and slow cooling rate of the molten pool were conducive to the floating of bubbles and avoiding porosity defects. The presence of a protective atmosphere and the non-existence of a filler metal reduced the impurities in the molten pool, which effectively prevented the formation of cracks.

## 4. Conclusions

Invar 36 alloy plates that were 10 mm thick were successfully welded using K-TIG welding with a single pass. The influence of heat input on the weld appearance, microstructure and properties was thoroughly investigated. The main conclusions were drawn as follows:With the increase in the heat input, the width of the weld root and reinforcement increased, while the depression increased.The grain size in both the HAZ and FZ increased with the increase in heat input. There was no change in the crystal structure across the joint, although texture changes occurred due to the weld thermal cycle, as observed by synchrotron X-ray diffraction.The tensile strength of the welded joints ranged between 428 and 438 MPa, while the elongation remained between 29.2 and 31.0%, indicating that the heat input had a low impact on the macroscopic tensile response of the Invar 36 joints.With the increase in heat input, the impact properties of both the HAZ and FZ in the welds decreased.The CTE of the welded joint was similar to that of the BM. No potential welding defects were observed in the joints for all selected heat inputs.

## Figures and Tables

**Figure 1 materials-16-03692-f001:**
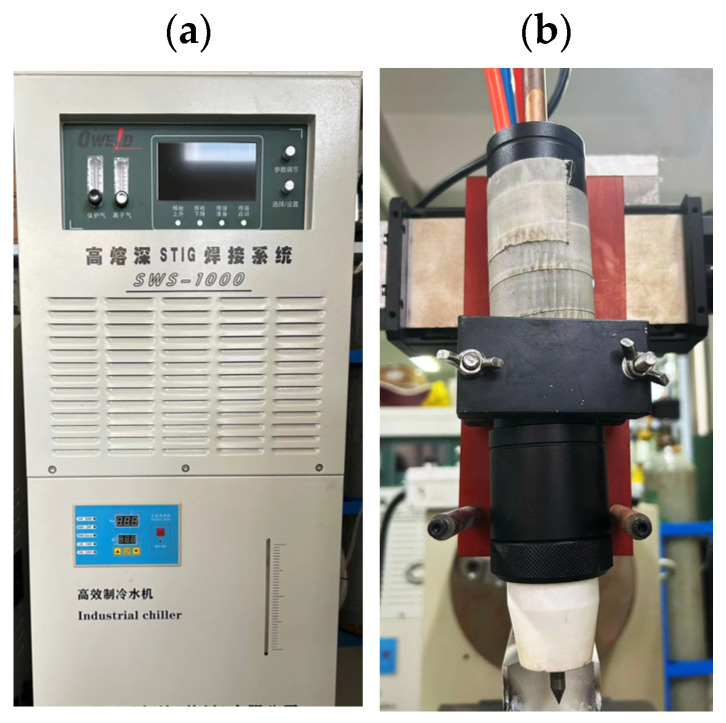
K-TIG welding equipment: (**a**) welding power system and (**b**) welding torch.

**Figure 2 materials-16-03692-f002:**
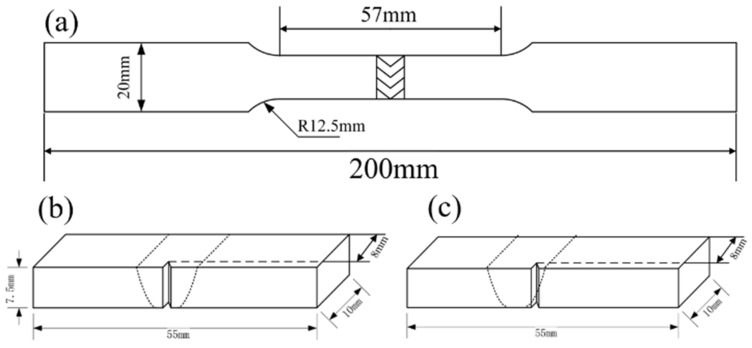
Dimensions of specimens for mechanical testing: (**a**) tensile specimens, (**b**) Charpy V-notch impact tests for the fusion zone, and (**c**) Charpy V-notch impact test for the heat affected zone.

**Figure 3 materials-16-03692-f003:**
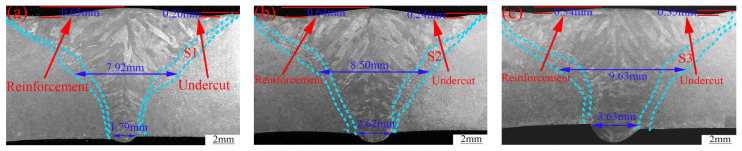
Cross-section of welds: (**a**) heat input of 2895.3 J/mm, (**b**) heat input of 3102.1 J/mm, and (**c**) heat input of 3308.9 J/mm. Close-up of the microstructure of the welds: (**d**) heat input of 2895.3 J/mm, (**e**) heat input of 3102.1 J/mm, and (**f**) heat input of 3308.9 J/mm.

**Figure 4 materials-16-03692-f004:**
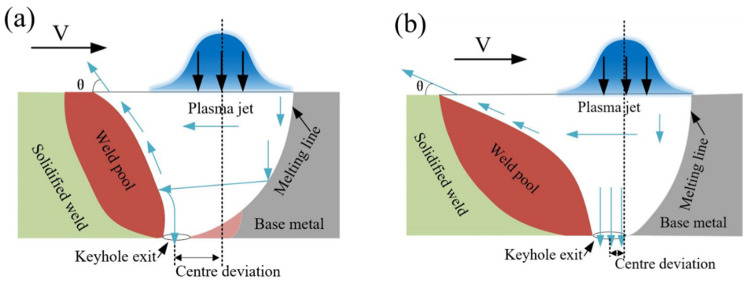
Schematic representation of the weld longitudinal cross-section and arc shape considering different heat inputs: (**a**) low heat input, and (**b**) high heat input.

**Figure 5 materials-16-03692-f005:**
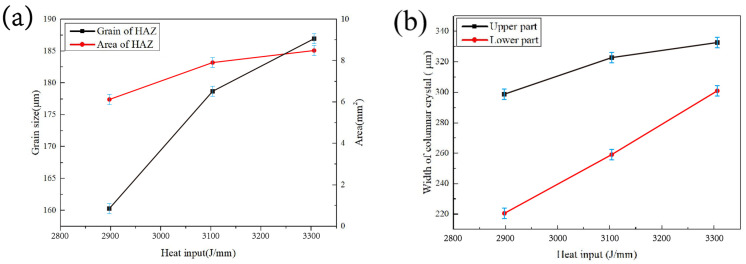
Influence of heat input on grain size: (**a**) area and grain size of HAZ, and (**b**) width of columnar grain in the FZ.

**Figure 6 materials-16-03692-f006:**
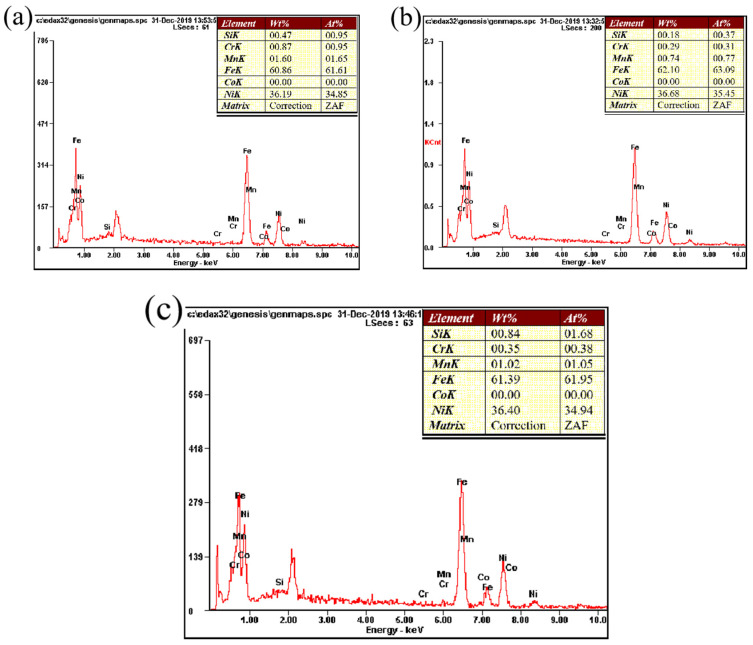
Composition of the FZ under different heat inputs: (**a**) spectrum 1 (2895.3 J/mm), (**b**) spectrum 2 (3102.1 J/mm), and (**c**) Spectrum3 (3308.9 J/mm).

**Figure 7 materials-16-03692-f007:**
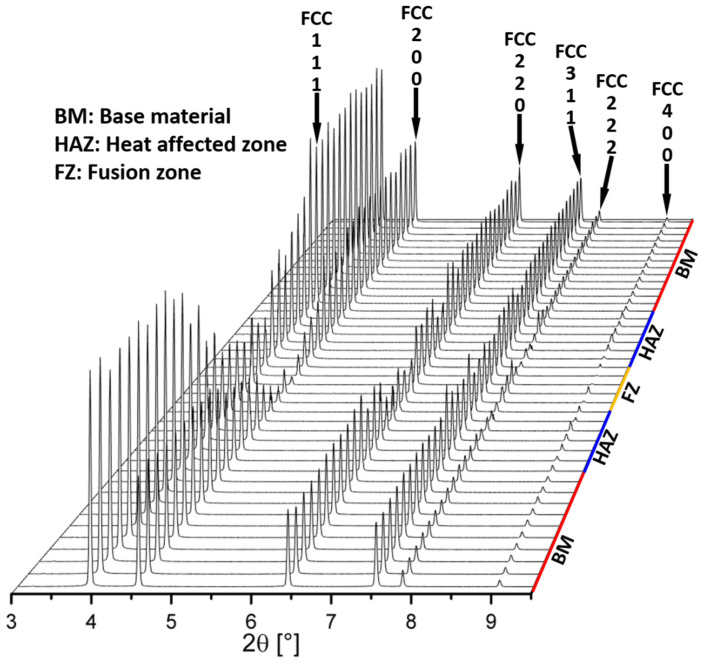
Superimposed synchrotron X-ray diffraction patterns across the welded joint (covering the BM, HAZ and FZ) of a K-TIG weld.

**Figure 8 materials-16-03692-f008:**
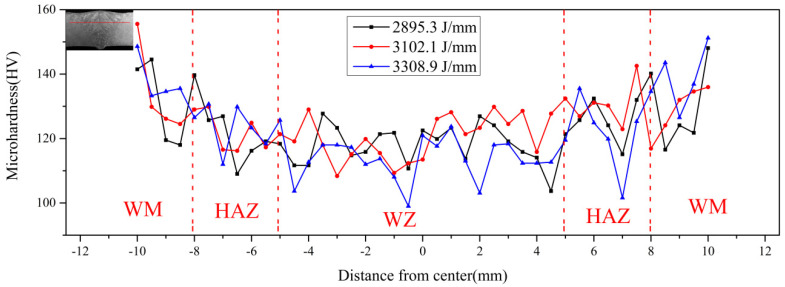
Microhardness of K-TIG welded joints for different heat inputs.

**Figure 9 materials-16-03692-f009:**
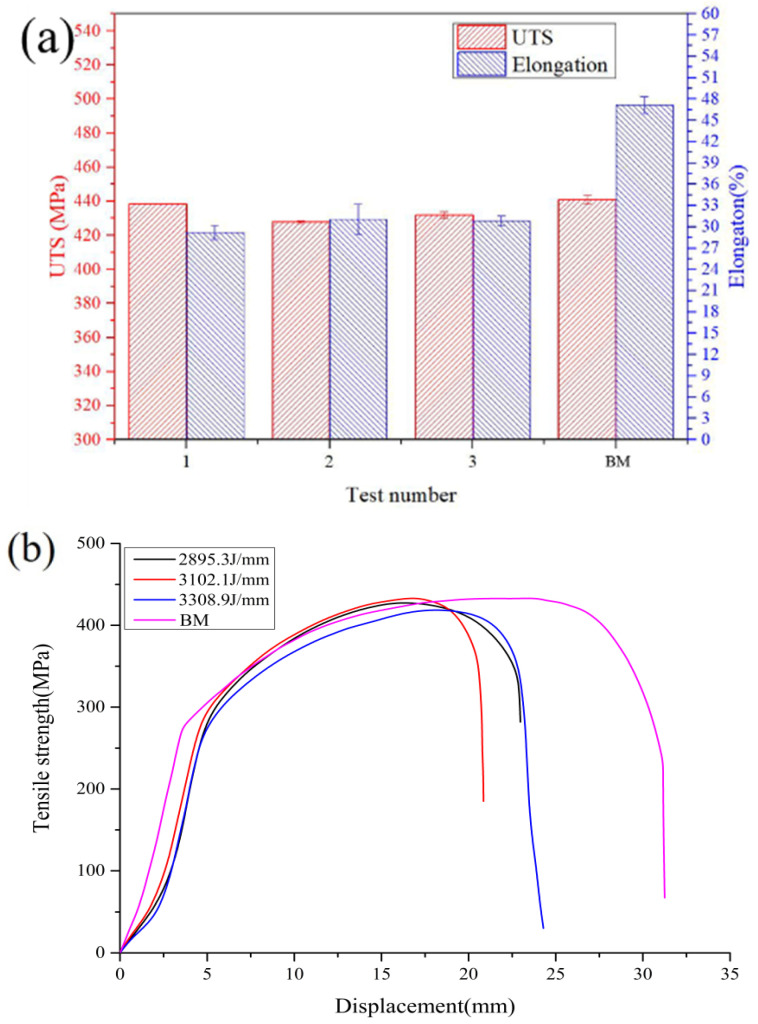
Tensile properties for K-TIG welds with different heat inputs: (**a**) average UTS and elongation, and (**b**) representative tensile curves.

**Figure 10 materials-16-03692-f010:**
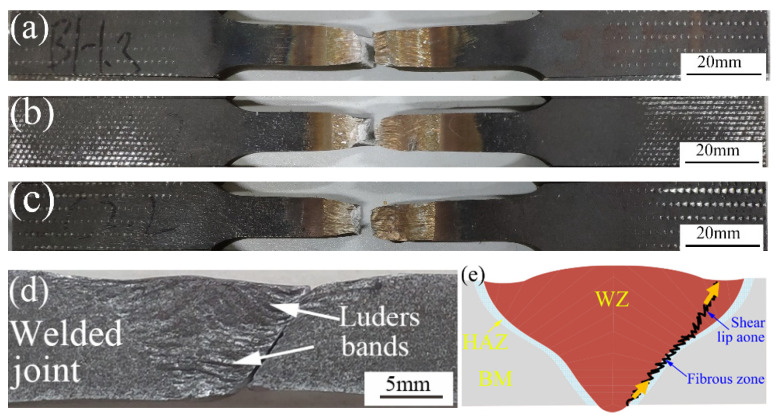
Fractured specimens of the K-TIG welds: (**a**) Sample 1 (2895.3 J/mm), (**b**) Sample 2 (3102.1 J/mm), (**c**) Sample 3 (3308.9 J/mm), (**d**) side view of K-TIG, weld and (**e**) schematic illustration of fracture location.

**Figure 11 materials-16-03692-f011:**
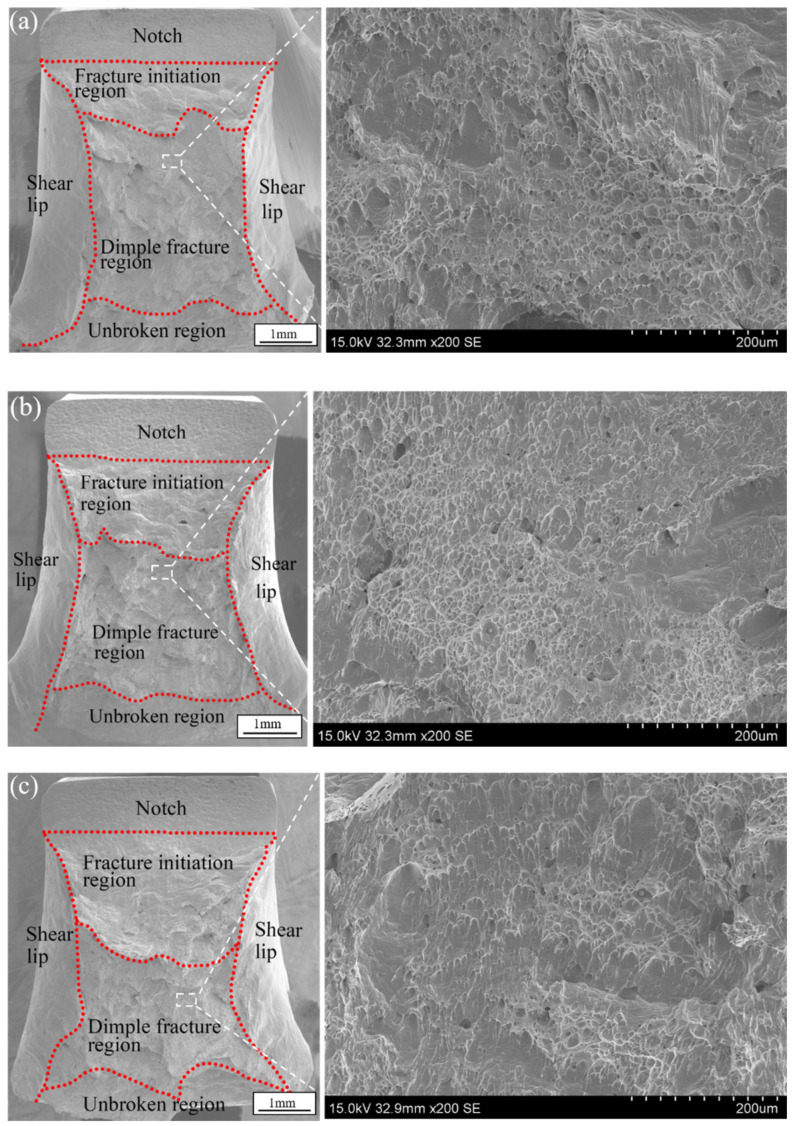
Fracture morphology of the notch in the FZ: (**a**) Sample 1 (2895.3 J/mm), (**b**) Sample 2 (3102.1 J/mm), and (**c**) Sample 3 (3308.9 J/mm).

**Figure 12 materials-16-03692-f012:**
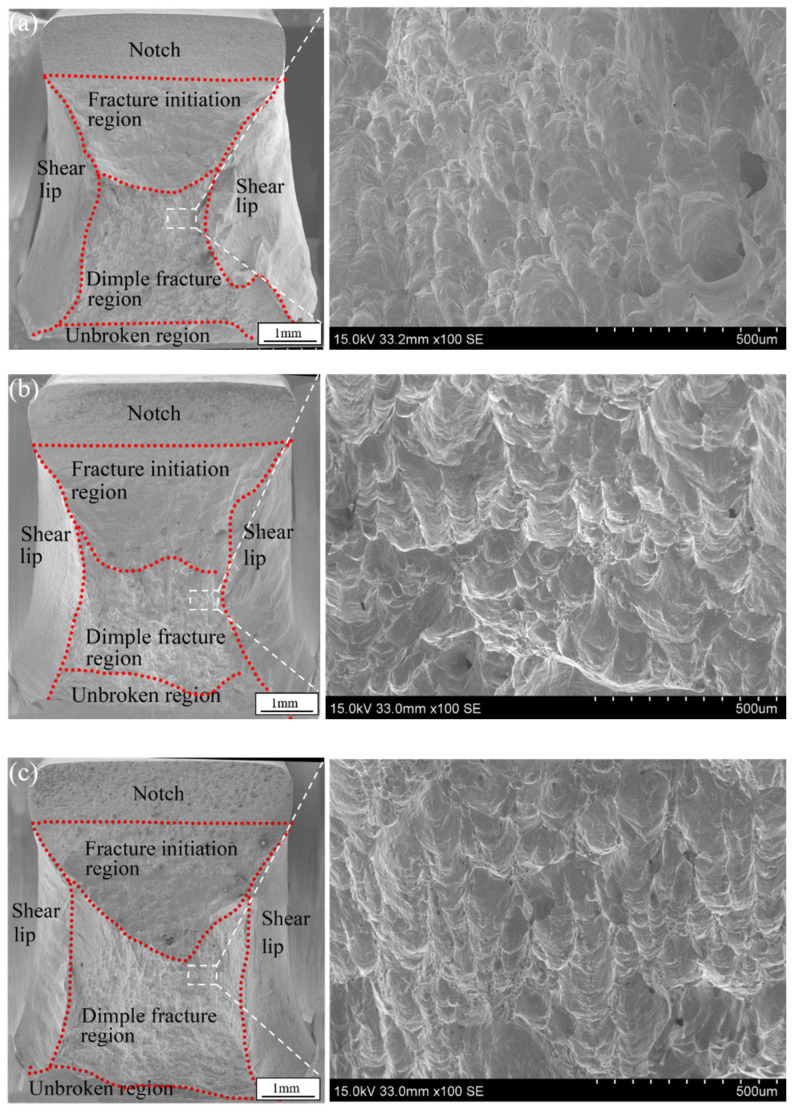
Fracture morphology of the notch in the HAZ: (**a**) Sample 1 (2895.3 J/mm), (**b**) Sample 2 (3102.1 J/mm) and (**c**) Sample 3 (3308.9 J/mm).

**Figure 13 materials-16-03692-f013:**
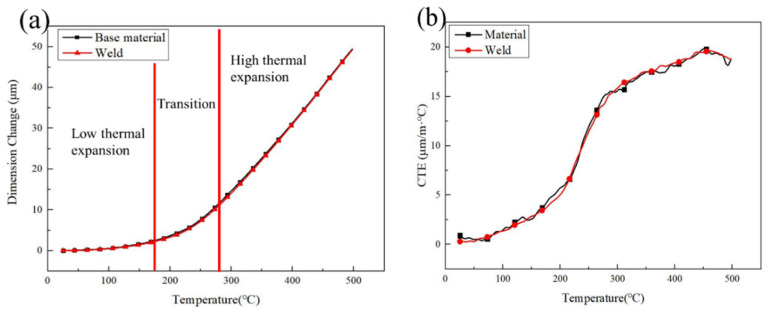
Variation in the CTE of Invar 36 alloy BM and welded joints: (**a**) sample elongation-temperature curve and (**b**) CTE vs. temperature curve.

**Figure 14 materials-16-03692-f014:**
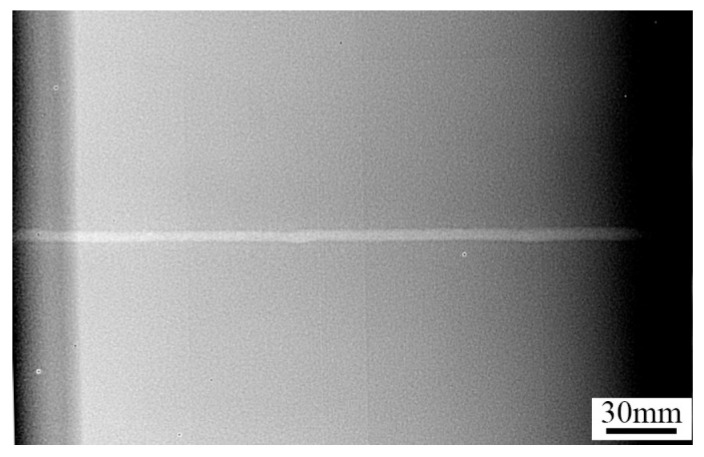
Radiography testing of the welded joint (heat input of 3102.1 J/mm).

**Table 1 materials-16-03692-t001:** Chemical composition of Invar 36 alloy (wt.%).

C	Si	Mn	P	S	Ni	Fe
0.05	0.2	0.2–0.6	0.02	0.02	35–37	Balance

**Table 2 materials-16-03692-t002:** Welding parameters for K-TIG welding of Invar 36 alloy.

Sample Number	Welding Current (A)	Welding Speed (mm/min)	Welding Voltage (V)	Heat Input (J/mm)	Joint Gap	Shielding Gas Flow Rate
1	420	2.35	18	2895.3	0	24 L/min
2	450	2.35	18	3102.1	0	24 L/min
3	480	2.35	18	3308.9	0	24 L/min

**Table 3 materials-16-03692-t003:** Impact properties of welds obtained with different heat inputs.

Sample Number	Heat Input (J/mm)	Impact Property, Ak (J)
FZ	HAZ
1	2895.3	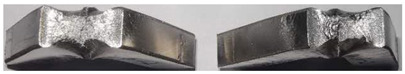
179.7	172.8
2	3102.1	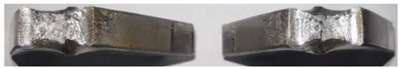
161.8	157.2
3	3308.9	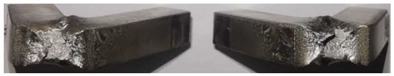
151.8	150.5
BM	---	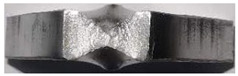
202.7

**Table 4 materials-16-03692-t004:** Comparison on the properties of Invar36 welded joints by K-TIG and other welding processes.

Welding Process	Thickness of BM (mm)	Weld Speed (mm/min)	Elongation (%)	Microhardness (HV)	UTS of BM (Ta) (MPa)	UTS of Weld (Tb) (MPa)	Tb/Ta (%)
K-TIG (present work)	10	235	29.2	115	441.0	438.0	99.3
FSW [30]	3	120	22	144	516.0	472.0	91.5
Laser welding [31]	3	1200	12	130	517.6	440.0	85.1
Multi-layer multi-pass MIG welding [32]	19.05	27	38	138	432	363	84.0

## Data Availability

Not applicable.

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
