# Peer review of "Effects of Heat Input on Weld Microstructure and Properties in Keyhole TIG Welding of Invar 36 Alloy"

_materials, 2023, doi:10.3390/ma16103692_

Round 1

Reviewer 1 Report

Authors studied the effect of heating on the microstructure and properties of Invar 36 alloy Welded using K-TIG welding. Results are encouraging. No potential welding defects were observed in the joints. In the following, few points need to be addressed:

1) Line 32: “Invar alloys have been widely used in large moulds in aerospace industry and liquefied natural gas tankers”. Please add references here.

2) Line 68:  “…was applied to butt joining of 10-mm-thick Invar 36 alloy plates”: Please add full stop at the end of the sentence.

3) Lines 81-83: Please put the definition of each parameter in a separate line.

4) Line 104: Please delete the full stop just before the word “.Results”.

5) Lines 261-264: Please put the definition of each parameter in a separate line.

6) Lines 314: Please consider the following new sentence: “No potential welding defects were observed in the joints for all selected heat inputs.”

Reviewer 2 Report

This study presents K.-TIG welding with different heat input applied using a but joining in 10 mm thick plates. Thea authors analyzed the weld appearance and microstructure. Some weld properties were exlored in terms of hardness, tensile strength and impact properties. 

-The introduction is interesting in terms of the novelty of this treatment. Please add relevant information about a engineering comparative between welding technologies. 

-Please include the limitations of K-TIG welding. 

-Materials and methods: FIgure 1 is not clear, it could be the resolution. When I read your study it was clear in introduction the several techniques to study however in materials and methods section (Experimental) is not clear, please use subheading to each tecnique thatyou will report in results, indicating machine, steps, configurations, etc. 

-Figure 3 (d,e,f) do not have scales.

-Figure 3, There is not clearly identified the zone, please change the line color and text color., 

-Figure 4 depicts the schematic diagram of weld pool under different heat inputs. When the heat input was low, melting of material at the bottom of the melt pool was not complete, which led to the large bending of the front wall of the pool. A large amount of plasma was discharged upward to squeeze the molten pool, which contributed to the for mation of reinforcement. The increase of reinforcement may affect the tensile properties and fracture of the joint [22]. Please discuss your results in terms of reference 22 and Figure 4. 

-Please add a section of discussion with al your results (microstructure,  hardness, tensile strength and impact properties). If the literature is limitated please compare with other welding technologies as laser welding, MIG, etc. 

The results are very valuable however, it is missing the discussion of your results and the comparative with other authors. 

Reviewer 3 Report

The paper measures the impact of heat input during a butt welding of INVAR alloys using key hole TIG welding. The authors employ a range of techniques to obtain the variations in microstructure of FZ and HAZ and try to correlate the results to impact measurements. The authors see an increase in the HAZ and variations in grain size and texture with increased heat input which is correlated to the impact measurements. The paper is well written.

1. However, the reviewer does not think the authors have done a good job in drawing a correlation between the observed microstructure variations (small increase in grain size & columnar width), large texture variations to the observed decrease in impact parameters even when the other mechanical tests showed an overall reduction with respect to BM but not much difference between different welds with different heat inputs (which is a little confusing).  There is not much difference in alloy composition even with increase in heat input where some diffusion, precipitation or carbide formation could have been expected (not much variation either-surprising). The explanation for the observed reduction in impact parameters are solely put on grain size variation (when the percentage variation is small).

2. I am confused by higher grain sizes in FZ. Even though FZ receives higher heat input and subsequent melting, shouldn't there be faster thermal drop leading to insufficient time for large autenitic grain formation? Structures where martensitic phase exists usually one sees the needle structure? Why is it different for INVAR alloy (the referee is not well versed in phase diagram of this alloy and hence the question).

3. Some of the measurements lack error bar. Please put error wherever it is possible.

4. Finally, the referee understands the need for testing new welding schemes with alloys widely used in the industry. The manuscript does that effectively but fails to advance any new knowledge regarding the system or the weld technology used.

Reviewer 4 Report

Manuscript is devoted to study the influence of heat input on the weld appearance, microstructure properties of 10-mm-thick Invar 36 alloy plates. The authors used various experimental techniques for studying the structure and properties and obtained important results. The article may be published after correction.

Main notes:

1. The research methods used are simply listed without a detailed description: pameters used, methods of preparing specimens for research, etc. This should be added.

2. Some conclusions are based on the change in austenite grain size. These changes are not very large. How accurate was the grain size determined? There is no information about  the accuracy  of  the grain size determination. What is the magnitude of the measurement error?  

3. The same remark applies to hardness data. The hardness measurement error should be specified (Fig. 8). Without specifying the accuracy of changes, fig. 8 is completely uninformative.

4. There are minor errors in the text, for example,  extra points (2. . Experimental)

Round 2

Reviewer 3 Report

Authors have addressed all concerns and hence the revised manuscript can be accepted.